# In Vitro Propagation of *Clausena lenis* Drake

**DOI:** 10.3390/plants14071123

**Published:** 2025-04-04

**Authors:** Pajaree Sathuphan, Srunya Vajrodaya, Nuttha Sanevas, Narong Wongkantrakorn

**Affiliations:** Department of Botany, Faculty of Science, Kasetsart University, Bangkok 10900, Thailand; pajaree.sat@ku.th (P.S.); fscisyv@ku.ac.th (S.V.); fscintsv@ku.ac.th (N.S.)

**Keywords:** callus induction, disinfection, plant tissue culture, medicinal plant, plant growth regulators

## Abstract

*Clausena lenis* Drake, a valuable medicinal plant in the Rutaceae family, faces threats from wildlife predation, overharvesting, and climate change. In the wild, *C. lenis* primarily propagates through seeds; however, their rapid loss of viability poses challenges for long-term storage and germplasm conservation. Plant tissue culture offers a practical solution for both its conservation and large-scale production. This study examines seed sterilization, callus induction, shoot multiplication, and root induction protocols for *C. lenis*. Seeds attained a 100% sterilization rate using 0.2% (*w*/*v*) HgCl_2_ for 20 min without compromising germination. When cultured on MS medium containing 0.5 mg/L 2,4-D, seed, stem-node, and 1-week-old seedling explants produced abundant callus. A 2.0 mg/L BA treatment achieved 100% shoot induction, with stem-node explants yielding the highest shoot proliferation (3.90 ± 0.31 shoots/explant), followed by 1-week-old seedlings (2.30 ± 0.21 shoots/explant) and seed explants (1.60 ± 0.16 shoots/explant). Rooting was most effective on half-strength MS medium supplemented with 20.0 mg/L IBA, producing an average of 4.30 ± 0.83 roots per shoot in shoot-tip-deprived explants. The rooted plantlets successfully acclimatized, attaining a 100% survival rate in a 1:1:1 mixture of sterile soil, cocopeat, and vermiculite. These findings provide a robust platform for the sustainable propagation and conservation of *C. lenis* in response to its growing vulnerabilities.

## 1. Introduction

*Clausena lenis* Drake is a woody shrub in the Rutaceae family, recognized for its strong aroma and commonly found in mountainous forests at elevations of 500–1300 m across China, Laos, Vietnam, and northern Thailand. Various parts of *C. lenis*, including leaves, young branches, flowers, fruits, and seeds, produce aromatic essential oils [1]. *C. lenis* exhibits notable medicinal potential, with reported anti-inflammatory, antiviral, antioxidant, cytotoxic, and antibacterial activities [2]. Bioactive compounds such as furanocoumarins, carbazole alkaloids, and O-terpenoidal coumarins—including a novel dimeric coumarin and three new phenylpropanoids—have been identified in its aerial parts [3,4]. These constituents support the plant’s traditional use in herbal medicine and highlight its promise for pharmaceutical and nutraceutical applications [3]. Preliminary research also indicates that root bark extracts of *C. lenis* may inhibit human pathogenic bacteria [5]. However, ongoing threats from climate change and overharvesting have put *C. lenis* and other medicinal plants in Thailand at risk [6,7,8,9].

Aside from limited seed viability in wild populations, *C. lenis* also faces reproductive challenges under cultivation. These include slow growth, irregular flowering, and poor germination rate. Such reproductive constraints pose significant barriers to both sustainable cultivation and conservation of the species. Plant tissue culture technology presents a promising strategy for conserving *C. lenis* and other valuable species. By offering rapid and controlled multiplication of genetically uniform plantlets, tissue culture supports both large-scale sustainable production and conservation efforts [10,11,12]. Furthermore, callus cultures derived from *C. lenis* may serve as a rich source of secondary metabolites [13]. While tissue culture protocols have been explored in other *Clausena* species, such as *C. guillauminii* [14] and *C. harmandiana* [15], to the best of our knowledge, there have been no published reports on in vitro propagation protocols specifically for *C. lenis*. Successful micropropagation typically depends on factors such as effective disinfection methods, balanced nutrient media, and optimal plant growth regulators (especially auxins and cytokinins). A higher cytokinin-to-auxin ratio often favors shoot proliferation, whereas a higher auxin concentration encourages root formation [16]. In *Plumbago auriculata*, adjusting auxin and cytokinin levels significantly improved both callus induction and shoot regeneration [17]. In *Asparagus densiflorus*, callus growth was more rapid on MS medium containing 5.4 μM pCPA and 4.4 μM BA compared to other combinations. BA alone was more effective than kinetin for shoot regeneration, and the addition of ancymidol to 0.4 μM BA significantly increased shoot numbers [18]. In *Tectona grandis*, specific hormonal combinations also enhanced shoot and root numbers [19]. These outcomes demonstrate how tailored hormone regimes can optimize specific developmental stages in plant tissue culture depending on species and explant type.

Developing a reproducible and well-characterized in vitro propagation protocol for ***C. lenis*** is important for providing a consistent supply of healthy plants with high propagation potential. It also offers an avenue for producing high-value secondary metabolites under controlled conditions. This study aims to determine effective disinfection strategies for *C. lenis* seeds, optimize protocols for callus induction, shoot and root formation directly from explants, and evaluate the survival rate of regenerated *C. lenis* plantlets once transferred to ex vitro conditions.

## 2. Results

### 2.1. Seed Surface Sterilization of C. lenis

The effectiveness of sodium hypochlorite (NaClO) and mercuric chloride (HgCl_2_) treatments for seed surface sterilization in *C. lenis* varied significantly, influencing both the decontamination and germination rates (Table 1 and Figure 1). Lower NaClO concentrations (0.12–0.6% for 20 min), even when used in a two-step disinfection (NaClO followed by HgCl_2_), achieved comparatively low sterilization rates (0–50%). This result suggests that these concentrations and durations of NaClO were insufficient to eliminate bacterial contaminants fully. In contrast, seed treatments employing HgCl_2_ (0.1–0.2% for 10–20 min) showed higher sterilization rates (80–100%). The most notable outcome was achieved with 0.1% and 0.2% HgCl_2_ for 20 min, both yielding 100% germination without initial contamination. However, seeds treated with 0.1% HgCl_2_ for 20 min began showing contamination after 8 weeks in culture. Consequently, 0.2% HgCl_2_ for 20 min was identified as the most suitable treatment to ensure both high disinfection efficiency and retention of seed germination capacity. This adjusted protocol was utilized in subsequent stages of the study.

### 2.2. Effect of Plant Growth Regulators on Callus Induction

Table 2 and Figure 2 illustrate the influence of various plant growth regulators (PGRs) on callus induction in three different explant types (seeds, stem segments with nodes, and 1-week-old seedlings) over 3 months of culture. MS medium devoid of PGRs served as the control and yielded no callus formation. Conversely, media supplemented with 0.5 and 1.0 mg/L 2,4-D substantially enhanced callus production (70–100% callus induction across explant types), reflecting the efficacy of this auxin in promoting callus initiation. Among the explant types, seeds showed the highest callus induction rate (100%). This indicates that seed explants are a more responsive source for callus production in *C. lenis* compared to stem segments or 1-week-old seedlings.

### 2.3. Effect of Plant Growth Regulators on Shoot Induction

The impact of different PGRs on shoot induction from seeds, stem segments with four nodes, and 1-week-old seedlings without roots is presented in Table 3. Various concentrations of BA, 2,4-D, TDZ, and NAA were tested on seed explants, stem segments with nodes, and 1-week-old seedlings. Several treatments achieved 100% shoot induction; however, the final shoot number per explant differed across treatments. The highest shoot proliferation typically occurred in the presence of 2.0 mg/L BA. Seed explants produced an average of 1.60 ± 0.16 shoots (Figure 3A), while stem explants with nodes yielded 3.90 ± 0.31 shoots (Figure 3B), and 1-week-old seedlings produced 2.30 ± 0.21 shoots (Figure 3C).

### 2.4. Effect of Plant Growth Regulators on Root Induction

Table 4 details the influence of indole-3-butyric acid (IBA) and indole-3-acetic acid (IAA) on root development in *C. lenis* over a 5-week period. Shoots with and without a shoot tip (approximately 3 cm long) were tested on half-strength MS medium containing 0, 4.0, 12.0, or 20.0 mg/L of each auxin. IBA treatments consistently produced higher root induction rates compared to IAA, particularly in shoot explants (cuttings with four nodes) lacking a shoot tip. The largest number of roots per explant (4.30 ± 0.83) was observed in the treatment supplemented with 20.0 mg/L IBA (Figure 4C), followed by 12.0 mg/L IBA (3.50 ± 0.87; Figure 4B) and the auxin-free control (3.10 ± 0.69; Figure 4A). Although differences among these three treatments were not statistically significant, these results underscore the potential of IBA to enhance root formation in *C. lenis*, especially in shoot-tip-deprived explants.

### 2.5. Plantlet Acclimatization

After rooting, well-developed plantlets were transferred to small plastic pots. Three different potting substrates were tested for acclimatization: sterile soil, a sterile soil–cocopeat mixture (1:1, *v*/*v*), and a sterile soil–cocopeat–vermiculite mixture (1:1:1, *v*/*v*/*v*) (Figure 5). Transparent plastic covers were used initially to maintain high humidity for 2 weeks. Following cover removal and an additional 2 weeks of growth, the survival rates were recorded. While both the sterile soil and the soil–cocopeat mixture yielded 80% survival, the soil–cocopeat–vermiculite combination supported the highest survival (100%). This indicates that a well-aerated substrate with balanced moisture retention is beneficial for the successful acclimatization of *C. lenis* plantlets.

## 3. Discussion

The success of surface sterilization for seed and explants is influenced by various factors, including disinfectant concentration, duration of exposure, and the sensitivity of explants to chemical agents. Achieving high disinfection rates without compromising seed viability is essential for subsequent tissue culture steps [20,21]. In this study, 0.2% HgCl_2_ for 20 min was found as the most effective protocol for balancing sterility and germination, aligning with previous work indicating that mercuric chloride can outperform sodium hypochlorite in several species [22,23,24,25,26]. Although HgCl_2_ poses environmental and safety concerns, it was employed here due to the ineffectiveness of NaClO-based protocols in achieving contamination-free germination.

Subsequent experiments focused on the effects of various plant growth regulators on callus and shoot induction in three explant types (seeds, stem with nodes, and 1-week-old seedlings) cultured on MS media supplemented with BA, 2,4-D, TDZ, and NAA. Our results indicated that 2,4-D played a pivotal role in callus induction, particularly at 0.5 mg/L with seed explants displaying the highest induction rate (100%), consistent with its known function in promoting cell division and dedifferentiation [27]. This aligns with studies showing that younger tissues, such as seeds, frequently exhibit increased responsiveness to auxins, such as Himalayan rice [28]. Nonetheless, careful optimization of 2,4-D levels is crucial, as excessive amounts can inhibit callus growth or produce abnormal morphogenesis [29,30,31]. Chitdacha (2018) [32] demonstrated that 0.5 mg/L 2,4-D resulted in high-frequency callus induction in *Moringa oleifera*, Moosikapala (2001) [33] found that 1.0 mg/L 2,4-D had a significant effect on callus induction percentage in *Garcinia mangostana* and 1.0 mg/L 2,4-D in combination with 0.5 mg/L BA being the most effective for callus induction in *Garcinia dulcis*, and Soorni (2015) [34] showed that 2,4-D had a significant effect on callus induction percentage in cumin. Given its effectiveness and frequent use in callus induction protocols, 2,4-D was employed at minimal concentrations to balance efficacy with safety. Furthermore, the use of different PGRs and their combinations significantly influences callus formation in various plant species. For example, Kitisripanya (2020) [13] found that 0.1 mg/L TDZ in combination with 1 mg/L NAA was effective in callus induction in *Clausena harmandiana*. Kanwar (2018) [35] found that a combination of 2,4-D and NAA resulted in high callus induction rates in *Dianthus caryophyllus* L. Kongkaew (2016) [36] reported that 2.0 mg/L 2,4-D and 0.5 mg/L BA, when combined, led to the highest callus in *Hevea brasiliensis*. These studies collectively highlight the importance of PGRs and their combinations in influencing callus formation and regeneration in various plant species.

Shoot induction results demonstrated that BA markedly enhanced shoot formation, particularly in stem explants with nodes. These nodal segments likely possess meristematic tissues in axillary buds that are primed for shoot development under optimal cytokinin levels [37]. In this study, 2.0 mg/L BA alone was sufficient to induce 100% shoot formation, with an average of 3.90 ± 0.31 shoots per explant. This supports earlier findings in *Asparagus densiflorus* and *Terminalia bellerica* using BA alone [18,38]. While other studies used combined PGRs (e.g., BA with TDZ or auxins) [29,39], our results suggest BA by itself can be effective for shoot induction in *C. lenis*.

Regarding root induction, half-strength MS medium supplemented with 20.0 mg/L IBA proved the most effective, particularly in shoots deprived of a shoot tip. Removing the shoot tip appears to reduce apical dominance, thereby redistributing auxin and promoting root meristem activity [40]. These results mirror those in *Conocarpus erectus*, *Citrus aurantifolia*, and *Citrus limettioides*, where IBA significantly improved rooting outcomes [41,42,43]. The high concentration of IBA (20.0 mg/L) likely stimulated robust root formation, yet further studies could explore whether slightly lower concentrations might achieve comparable results with reduced input costs or phytotoxic risks.

In the final stage, plantlets with healthy roots survived at rates up to 100% when transferred to a 1:1:1 mixture of sterile soil, cocopeat, and vermiculite. This outcome aligns with research on papaya (*Carica papaya* L.) [44] and peach (*Prunus persica*) [45], where vermiculite’s superior aeration and moisture-retention properties promoted successful acclimatization. Hence, the substrate choice proved vital for minimizing transplant shock and ensuring stable ex vitro establishment.

Collectively, these findings provide a comprehensive tissue culture platform for *C. lenis*. Effective seed sterilization, suitable PGR combinations for callus and shoot induction, and targeted root induction protocols shown here present a viable approach to conserving this medicinally and ecologically significant species. Moreover, high survival rates during acclimatization support the method’s potential for scaling up propagation and ensuring the long-term sustainability of *C. lenis*.

## 4. Materials and Methods

### 4.1. Plant Materials

Mature fruits of *C. lenis* were collected from Chiang Dao District, Chiang Mai Province, Thailand. The pulp was removed to obtain the seeds, which were thoroughly washed under running tap water with a mild detergent, followed by rinsing in tap water. The cleaned seeds were then air-dried before further processing.

### 4.2. Seed Surface Sterilization

The air-dried seeds of *C. lenis* were used as explants and surface-sterilized under aseptic conditions. First, they were immersed in 70% (*v*/*v*) ethanol for 1 min, and then 12 disinfection treatments were evaluated, varying in chemical agents, concentrations, and exposure durations (Table 1). NaClO was tested at five concentrations (0.12%, 0.3%, 0.6%, 0.9%, and 1.2%) for 20 min as single-step treatments. Additionally, three concentrations (0.6%, 0.9%, and 1.2%) were tested in combination with a second disinfection step using 0.1% mercuric chloride (HgCl_2_) for 5 min. Additional treatments included single-step disinfection with HgCl_2_ at two concentrations (0.1% and 0.2%) for either 10 or 20 min. Each treatment involved 10 seeds (one per bottle). Subsequently, the seeds were rinsed three times with sterile distilled water. The seed coats were removed before placing the seeds on Murashige and Skoog (MS) medium supplemented with 3% (*w*/*v*) sucrose and 0.82% (*w*/*v*) agar, sterilized by autoclaved at 121 °C for 15 min. Cultures were incubated under a 16 h photoperiod (130 μmol m^−2^ s^−1^) provided by cool-fluorescent lights (Thai Toshiba Lighting Co., Ltd., Pathum Thani, Thailand) for 2 weeks. The most effective sterilization protocol identified here was used in subsequent experiments. The disinfection rate is calculated by dividing the number of sterile seeds by the total number of seeds and then multiplying by 100. The germination rate is calculated by dividing the number of germinated seeds by the total number of seeds and then multiplying by 100.

### 4.3. Effects of Plant Growth Regulators on In Vitro Culture of C. lenis

Three types of sterile explants, including seeds, stem segments with four nodes (1.5–2.0 cm in length) taken from 1-month-old seedlings, and 1-week-old seedlings without roots, were obtained from germinated seeds of *C. lenis* and cultured on MS medium. The explants were cultured on MS medium supplemented with various concentrations of N^6^-Benzyladenine (BA), 2,4-Dichlorophenoxyacetic acid (2,4-D), Thidiazuron (TDZ), and α-Naphthalene acetic acid (NAA) at concentrations ranging from 1.0 to 2.0 mg/L for BA, 0.5 to 1.0 mg/L for 2,4-D, 0.1 mg/L for TDZ, and 1.0 to 2.0 mg/L for NAA, either individually or in combination, as presented in Table 2 and Table 3. Each treatment was performed once using 10 explants (n = 10). Cultures were maintained at 25 ± 2 °C under a 16 h photoperiod for 3 months. Data on callus induction, multiple shoot induction frequency, number of shoots per explant, and the optimal plant growth regulator combinations were recorded. The callus induction rate is calculated by dividing the number of explants that produced callus by the total number of explants and then multiplying by 100. The multiple shoot frequency rate is calculated by dividing the number of explants that produced multiple shoots by the total number of explants and then multiplying by 100.

### 4.4. Root Induction

Individual shoots (with or without a shoot tip) approximately 3 cm in length were separated from the shoot clumps and transferred to half-strength MS medium containing indole-3-acetic acid (IAA) or indole-3-butyric acid (IBA) at 0, 4.0, 12.0, or 20.0 mg L^−1^ (Table 3). Each treatment was performed once using 10 explants (n = 10), with one explant per bottle. Cultures were maintained at 25 ± 2 °C under a 16 h photoperiod. Root formation was monitored for 6 weeks, and growth parameters were recorded for statistical analysis.

### 4.5. Plantlet Acclimatization

Plantlets with well-developed roots were gently removed from the culture bottles, thoroughly rinsed under tap water to eliminate any residual agar and transferred to small plastic pots. Three substrate mixtures were tested: sterile soil, a 1:1 mixture of sterile soil and cocopeat, and a 1:1:1 mixture of sterile soil, cocopeat, and vermiculite. Transparent plastic covers were placed over the pots to maintain high humidity for 2 weeks. The covers were then removed, and the plantlets were grown under these conditions for another 2 weeks before exposure to direct sunlight. The percentage survival rate was determined at the end of the acclimatization period.

### 4.6. Data Analysis

Data analysis was performed using R Statistical Programming Version 4.3.3 [46]. As the residuals were not normally distributed, the Kruskal–Wallis test was used to assess differences among the treatments, followed by Fisher’s least significant difference (LSD) test with Holm’s method for *p*-value adjustment in post hoc comparisons.

## 5. Conclusions

This study establishes an effective stepwise protocol for the in vitro propagation of *C. lenis* using three explant types: seeds, stem segments with nodes, and 1-week-old seedlings. For disinfection, treating seeds with 0.2% (*w*/*v*) HgCl_2_ for 20 min resulted in 100% sterilization and germination, making it the most effective sterilization protocol tested. Callus induction was most successful when seed explants were cultured on full-strength MS medium supplemented with 0.5 mg/L 2,4-D, resulting in a 100% callus formation rate. For shoot multiplication, stem segments with nodes cultured on MS medium containing 2.0 mg/L BA achieved 100% shoot induction, producing an average of 3.90 ± 0.31 shoots per explant. Seed explants and 1-week-old seedlings under the same treatment showed lower shoot proliferation (1.60 ± 0.16 and 2.30 ± 0.21 shoots per explant, respectively). For root induction, shoot explants without a shoot tip responded best when cultured on half-strength MS medium supplemented with 20.0 mg/L IBA, resulting in 90% rooting and an average of 4.30 ± 0.83 roots per explant. Finally, acclimatized plantlets showed 100% survival when transferred to a 1:1:1 mixture of sterile soil, cocopeat, and vermiculite. These results present a reliable micropropagation protocol for *C. lenis*, which may support its conservation, sustainable utilization, and future biotechnological applications.

## Figures and Tables

**Figure 1 plants-14-01123-f001:**
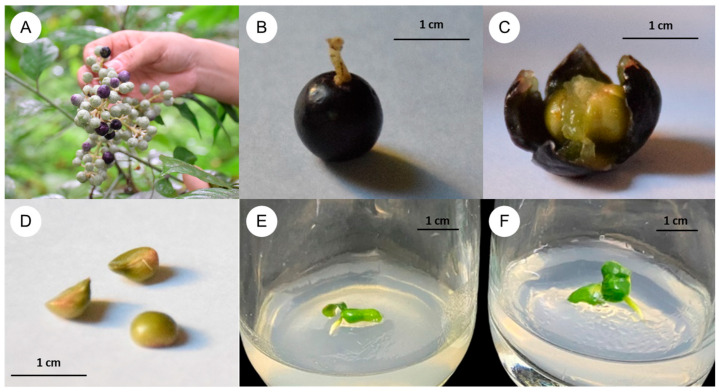
Effects of different disinfection methods on the seeds of *C. lenis*: (**A**–**C**) fresh mature fruits, (**D**) the seeds, (**E**) seedlings from seeds disinfected with 0.1% HgCl_2_, and (**F**) 0.2% HgCl_2_ for 20 min. after culture for 2 weeks.

**Figure 2 plants-14-01123-f002:**
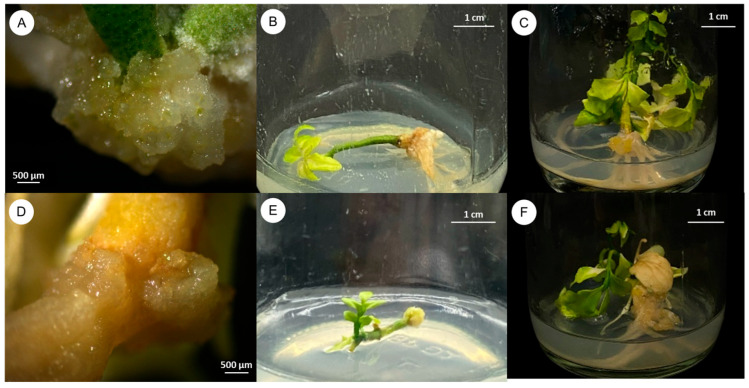
Effect of plant growth regulators on callus induction of *C. lenis* after culture for 3 months. (**A**) Seed explant cultured on MS with 0.5 mg/L 2,4-D, (**B**) stem with nodes explant cultured on MS with 0.5 mg/L 2,4-D, (**C**) 1-week-old shoot explant cultured on MS with 0.5 mg/L 2,4-D, (**D**) seed explant cultured on MS with 1.0 mg/L 2,4-D, (**E**) stem with nodes explant cultured on MS with 1.0 mg/L 2,4-D, and (**F**) 1-week-old shoot explant cultured on MS with 1.0 mg/L 2,4-D.

**Figure 3 plants-14-01123-f003:**
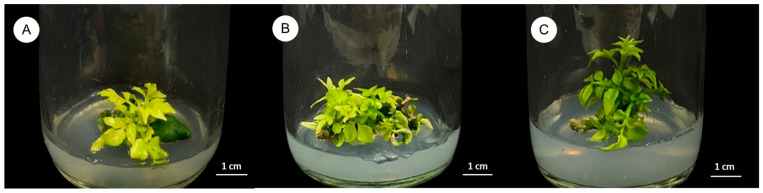
Effects of PGRs on shoot induction from different explants of *C. lenis* after culture for 3 months. (**A**) Seed explant cultured on MS with 2.0 mg/L BA, (**B**) stem with nodes explant cultured on MS with 2.0 mg/L BA, and (**C**) 1-week-old shoot explant cultured on MS with 2.0 mg/L BA.

**Figure 4 plants-14-01123-f004:**
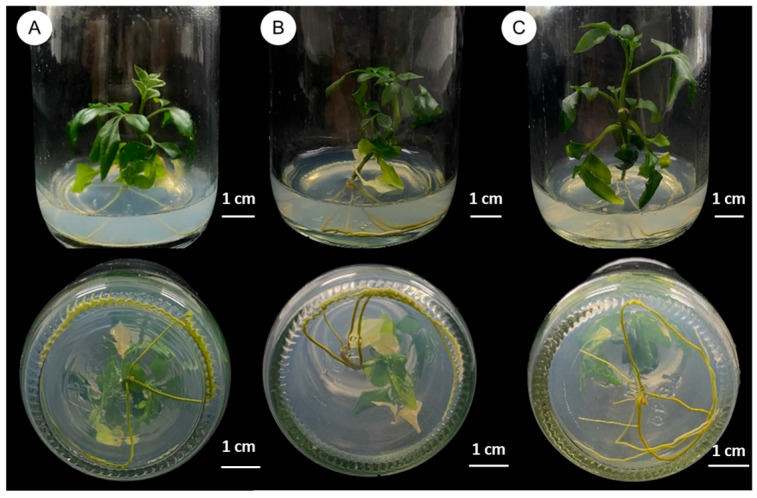
Effects of PGRs on root induction from shoot without shoot tip of *C. lenis* after culture for 6 weeks: (**A**) root induction on half-strength MS without PGRs, (**B**) root induction on half-strength MS with 12.0 mg/L IBA, and (**C**) root induction on half-strength MS with 20.0 mg/L IBA.

**Figure 5 plants-14-01123-f005:**
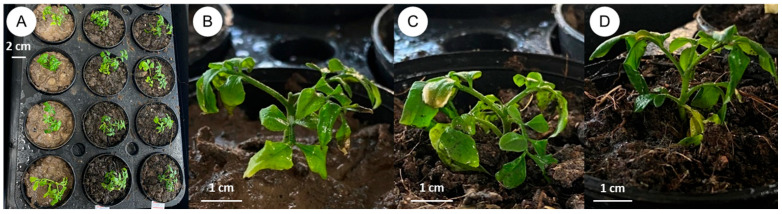
Plantlet acclimatization of *C. lenis*: (**A**) Plantlets with a well-developed root system are planted in small plastic pots filled with transplanting material on day one, (**B**) transplanted with sterile soil for 4 weeks, (**C**) transplanted with mixture of sterile soil and cocopeat (1:1) for 4 weeks, and (**D**) transplanted with mixture of sterile soil, cocopeat, and vermiculite (1:1:1) for 4 weeks.

**Table 1 plants-14-01123-t001:** Effects of different disinfection methods on the seeds of *C. lenis.* Means with same letter(s) in a column are not significantly different by Fisher’s least significant difference (LSD) test with the Holm’s method for *p*-adjustment at *p* ≤ 0.05. Asterisks indicate the significant differences.

Treatments	1st Disinfection	2nd Disinfection	Disinfection Rate (%)	Germination Rate (%)
1	0.12% NaClO (20 min)	-	0 c	80.00 ± 13.33 b
2	0.3% NaClO (20 min)	-	0 c	100.00 ± 0.00 a
3	0.6% NaClO (20 min)	-	0 c	100.00 ± 0.00 a
4	0.9% NaClO (20 min)	-	20.00 ± 13.33 c	100.00 ± 0.00 a
5	1.2% NaClO (20 min)	-	10.00 ± 10.00 c	100.00 ± 0.00 a
6	0.6% NaClO (20 min)	0.1% HgCl_2_ (5 min)	10.00 ± 10.00 c	100.00 ± 0.00 a
7	0.9% NaClO (20 min)	0.1% HgCl_2_ (5 min)	40.00 ± 16.33 bc	100.00 ± 0.00 a
8	1.2% NaClO (20 min)	0.1% HgCl_2_ (5 min)	50.00 ± 16.67 abc	100.00 ± 0.00 a
9	0.1% HgCl_2_ (10 min)	-	80.00 ± 13.33 ab	100.00 ± 0.00 a
10	0.1% HgCl_2_ (20 min)	-	100.00 ± 0.00 a	100.00 ± 0.00 a
11	0.2% HgCl_2_ (10 min)	-	80.00 ± 13.33 ab	100.00 ± 0.00 a
12	0.2% HgCl_2_ (20 min)	-	100.00 ± 0.00 a	100.00 ± 0.00 a
		Kruskal-Wallis test	<0.001 *	<0.001 *

**Table 2 plants-14-01123-t002:** Effects of PGRs on callus induction from different explants of *C. lenis*. Means having same letter(s) in a column are not significantly different by Fisher’s least significant difference (LSD) test with the Holm’s method for *p*-adjustment at *p* ≤ 0.05. Asterisks indicate the significant differences.

Treatments	PGRs (mg/L)				Callus Induction (%)		
	BA	2,4-D	TDZ	NAA	Seed	Stem with Nodes	1-Week-Old Shoot
**1**	0	0	0	0	0 c	0 b	0 c
**2**	0	0.5	0	0	100.00 ± 0.00 a	70.00 ± 15.28 a	80.00 ± 13.33 a
**3**	0	1.0	0	0	100.00 ± 0.00 a	40.00 ± 16.33 ab	90.00 ± 10.00 a
**4**	1.0	0	0	0	0 c	0 b	0 c
**5**	1.0	0.5	0	0	70.00 ± 15.27 ab	30.00 ± 15.28 ab	0 c
**6**	1.0	1.0	0	0	50.00 ± 16.67 b	40.00 ± 16.33 ab	40.00 ± 16.33 b
**7**	2.0	0	0	0	0 c	0 b	0 c
**8**	2.0	0.5	0	0	0 c	0 b	20.00 ± 13.33 bc
**9**	2.0	1.0	0	0	100.00 ± 0.00 a	60.00 ± 16.33 ab	0 c
**10**	0	0	0	1.0	60.00 ± 16.33 b	50.00 ± 16.67 ab	0 c
**11**	0	0	0	2.0	100.00 ± 0.00 a	0 b	0 c
**12**	0	0	0.1	0	0 c	0 b	0 c
**13**	0	0	0.1	1.0	0 c	40.00 ± 16.33 ab	0 c
**14**	0	0	0.1	2.0	0 c	30.00 ± 15.28 ab	0 c
Kruskal-Wallis test					<0.001 *	<0.001 *	<0.001 *

**Table 3 plants-14-01123-t003:** Effects of PGRs on shoot induction from different explants of *C. lenis*. Means having same letter(s) in a column are not significantly different by Fisher’s least significant difference (LSD) test with the Holm’s method for *p*-adjustment at *p* ≤ 0.05. Asterisks indicate the significant differences.

Treatments	PGRs (mg/L)				Seed		Stem with Nodes		1-Week-Old Shoot	
	BA	2,4-D	TDZ	NAA	Multiple Shoot Frequency (%)	Number of Shoots per Explant	Multiple Shoot Frequency (%)	Number of Shoots per Explant	Multiple Shoot Frequency (%)	Number of Shoots per Explant
1	0	0	0	0	0	0 c	100	1.00 ± 0.00 f	0	0 e
2	0	0.5	0	0	0	0 c	100	1.00 ± 0.00 f	0	0 e
3	0	1.0	0	0	0	0 c	100	1.00 ± 0.00 f	0	0 e
4	1.0	0	0	0	80	1.30 ± 0.26 a	100	3.30 ± 0.21 ab	100	1.30 ± 0.15 ab
5	1.0	0.5	0	0	0	0 c	100	2.40 ± 0.31 bcd	10	0.10 ± 0.10 de
6	1.0	1.0	0	0	0	0 c	100	1.40 ± 0.16 ef	0	0 e
7	2.0	0	0	0	100	1.60 ± 0.16 a	100	3.90 ± 0.31 a	100	2.30 ± 0.21 a
8	2.0	0.5	0	0	0	0 c	100	3.00 ± 0.21 abcd	0	0 e
9	2.0	1.0	0	0	0	0 c	100	2.20 ± 0.25 cde	20	0.20 ± 0.13 de
10	0	0	0	1.0	0	0 c	100	1.00 ± 0.00 f	0	0 e
11	0	0	0	2.0	0	0 c	100	1.00 ± 0.00 f	0	0 e
12	0	0	0.1	0	40	0.40 ± 0.16 b	100	3.10 ± 0.18 abc	60	0.80 ± 0.25 bc
13	0	0	0.1	1.0	0	0 c	100	2.10 ± 0.28 de	50	0.50 ± 0.17 cd
14	0	0	0.1	2.0	20	0.20 ± 0.13 bc	100	2.60 ± 0.37 bcd	30	0.30 ± 0.15 cde
Kruskal-Wallis test					NA	<0.001 *	NA	<0.001 *	NA	<0.001 *

**Table 4 plants-14-01123-t004:** Effects of PGRs on root induction from different explants of *C. lenis*. Means with same letter(s) in a column are not significantly different by Fisher’s least significant difference (LSD) test with the Holm’s method for *p*-adjustment at *p* ≤ 0.05. Asterisks indicate the significant differences.

Treatments	Explants	PGRs (mg/L)		6 Weeks of Culture	
		IAA	IBA	Root Induction (%)	Number of Roots per Explant
**1**	Shoot with shoot tip	0	0	10	0.20 ± 0.20 b
**2**		4.0	0	0	0 b
**3**		12.0	0	0	0 b
**4**		20.0	0	0	0 b
**5**		0	4.0	20	0.70 ± 0.52 b
**6**		0	12.0	0	0 b
**7**		0	20.0	40	0.80 ± 0.36 b
**8**	Shoot withoutshoot tip	0	0	90	3.10 ± 0.69 a
**9**		4.0	0	10	0.20 ± 0.20 b
**10**		12.0	0	0	0 b
**11**		20.0	0	0	0 b
**12**		0	4.0	90	2.50 ± 0.48 a
**13**		0	12.0	80	3.50 ± 0.87 a
**14**		0	20.0	90	4.30 ± 0.83 a
Kruskal-Wallis test				NA	<0.001 *

## Data Availability

The original contributions presented in this study are included in the article. Further inquiries can be directed to the corresponding author.

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
