# Peer review of "In Vitro Propagation of Clausena lenis Drake"

_plants, 2025, doi:10.3390/plants14071123_

Round 1

Reviewer 1 Report

Comments and Suggestions for Authors

The manuscript is of average importance. It is typical paper investigating micropropagation protocol, using standard procedure for this kind of the research. Taking into account that Clausena lenis is useful medicinal plant, and that there is no propagation protocol available or the species, this research is important for its mass production. However, some parts of the manuscript can be improved. I inserted comments in the manuscript. 

Author Response

Page 1 Title: Modified from In Vitro Micropropagation to In Vitro Propagation

  • Modified accordingly

Page 2 line 53: Italicize Citrus medica

  • As we have removed the reference to Citrus medica, we reviewed the manuscript to ensure that all remaining scientific species names are properly italicized.

Page 6 line 143: change “shoot explant” to “the cuttings with four nodes”

  • We have modified this to “in shoot explants (cuttings with four nodes)”

Page 10 line 290: Does it mean 10 explants per treatment? How many repetitions were done?

  • Each treatment was performed once using 10 explants (n=10). We have added this sentence in Materials and Methods at P9L292 and P10L305.

Page 10 conclusion: The conclusions should contain sumarized protocol or micropropagation of this species with highlighted expected results. Informations about explant type, medium type and epected results should be provided. E.g.''The synthetic auxin 2,4-D (0.5 mg/L) significantly enhanced callus formation when /which explants/ were cultured on /which media/ with expected rate of callus formation of /percentage rom reults), while 2.0 mg/L BA proved highly effective for shoot induction, when /which explants/ were cultured on /which media/ resulting in /how many shoots/. Same or rooting phase, explanty type, the composition of medium as well as expected rooting percentage and number of roots should be mentioned.

We have revised the conclusion according to the reviewer’s suggestion. The updated version now includes a summarized stepwise protocol for micropropagation of Clausena lenis, detailing explant types, medium composition, and the expected outcomes for each stage of culture

Reviewer 2 Report

Comments and Suggestions for Authors

Title

In Vitro Micropropagation of Clausena lenis Drake

Abstract

Lines 8-9 have any reproductive problems?

Line 12: The use of HgCl2 should be the last option for disinfection, as it is harmful to the environment and human health.

Line 14: Likewise, the use of 2,4-D.

In general, the summary lacks a solid justification for conducting this study. Are there no previous reports?

Keywords

Do not repeat the words in the title.

Introduction

Lines 34-36: A citation is needed to support this idea.

Lines 32-41: The idea of ​​importance is the same (medicinal use): summarize.

Lines 42-44: Aside from this problem in wild populations, it would be good to explain whether the plant under study suffers from reproductive problems, justifying the use of PTC.

Line 48: What advantage does this study offer to those cited in this sentence?

Lines 53-55: What was obtained from these variations?

Line 53: What do you mean by an efficient protocol? It may already exist.

Lines 60-61: This is known as an indirect organogenesis micropropagation protocol.

Results

Table 1: Why wasn't NaCLO evaluated in 10 min?

Why wasn't a second disinfection used in NaCLO?

Should alternatives to using HgCl2 be sought?

Table 2: Why wasn't BAP vs. TDZ used?

Generating seed calluses could increase genetic variability.

Table 3: Wasn't shoot regeneration based on the generated calluses? What was the purpose of generating calluses?

The results are interesting; however, they are part of a routine study in PTC. Techniques applied to all plant species to be micropropagated.

Discussion

Lines 182-185: According to which author?

Lines 185-188, as can be seen, the use of HgCl2 in the cited works is not recent due to its current disuse.

Lines 188-190, perhaps with two NaClO rinses, the contamination results would have decreased, and the use of HgCl2 would be avoided.

Lines 190-193, in this study, seed conservation and storage were not evaluated; this statement is incorrect because it was not proven.

Lines 202-204, it would be worth discussing with another study where high doses of 2,4-D cause this.

Lines 209-210, according to whom?

Lines 221-225, in their study, BAP was not combined with TDZ; discuss comparable works on equal terms.

Materials and Methods

Line 260, details should be given in the text so that the methods can be reproducible.

Lines 261-263: Was the culture medium sterilized? How?

Line 271: Where did the three sources of explants come from? From germinated seeds? Specify.

Lines 274-276: What concentrations?

Lines 278: Why were they evaluated at three months? And in other sections at six weeks?

The materials and methods should be further detailed to make them reproducible.

Conclusion

Line 310: Conducting a study for the first time does not justify publishing it.

Lines 318-320: To meet these objectives, a conservation strategy should have been developed.

The conclusion makes statements that were not verified in this study and should be improved.

Author Response

Lines 8-9 have any reproductive problems?

- We added at P1L9-10: “In the wild, C. lenis primarily propagates through seeds, however, their rapid loss of viability poses challenges for long-term storage and germplasm conservation.”

Line 12: The use of HgCl2 should be the last option for disinfection, as it is harmful to the environment and human health.

- We acknowledge the concerns and now provide a justification in the discussion at P8L199 as follows: “Although HgCl₂ poses environmental and safety concerns, it was employed here due to the ineffectiveness of NaClO-based protocols in achieving contamination-free germination.”

Line 14: Likewise, the use of 2,4-D.

- We now address this in the discussion and clarify that 2,4-D was used at low concentrations, consistent with common practice in callus induction at P8L216 as follows: “Given its effectiveness and frequent use in callus induction protocols, 2,4-D was employed at minimal concentrations to balance efficacy with safety.”

In general, the summary lacks a solid justification for conducting this study. Are there no previous reports?

- We have revised the conclusions to better justify the rationale for this study. While tissue culture protocols have been explored in other Clausena species, such as C. excavata and C. indica, to the best of our knowledge, there have been no published reports on in vitro propagation protocols specifically for Clausena lenis. Given the increasing threats from overharvesting, habitat loss, and poor seed viability, developing a reliable micropropagation strategy for C. lenis is both timely and necessary.

Keywords

Do not repeat the words in the title.

- The word “micropropagation” and “Clausena lenis” were removed from the keywords. New keywords now include “callus induction”.

Introduction

Lines 34-36: A citation is needed to support this idea.

- A citation has been added to support the medicinal relevance of C. lenis at P1L35 as follows: “C. lenis exhibits notable medicinal potential, with reported anti-inflammatory, antiviral, antioxidant, cytotoxic, and antibacterial activities [2].”

Lines 32-41: The idea of ​​importance is the same (medicinal use): summarize.

- We condensed the paragraph to reduce redundancy and focus on core pharmacological properties at P1L35 as follows: “C. lenis exhibits notable medicinal potential, with reported anti-inflammatory, antiviral, antioxidant, cytotoxic, and antibacterial activities [2]. Bioactive compounds such as furanocoumarins, carbazole alkaloids, and O-terpenoidal coumarins—including a novel dimeric coumarin and three new phenylpropanoids—have been identified in its aerial parts [3,4]. These constituents support the plant’s traditional use in herbal medicine and highlight its promise for pharmaceutical and nutraceutical applications [3].”

Lines 42-44: Aside from this problem in wild populations, it would be good to explain whether the plant under study suffers from reproductive problems, justifying the use of PTC.

- We addressed this point by noting seed recalcitrance and difficulty in propagation under natural conditions at P1L44 as follows: “Aside from limited seed viability in wild populations, C. lenis also faces reproductive challenges under cultivation. These include slow growth, irregular flowering, and poor germination rate. Such reproductive constraints pose significant barriers to both sustainable cultivation and conservation of the species.”

Line 48: What advantage does this study offer to those cited in this sentence?

- While previous studies (e.g., Sudhersan et al., 2003; Hussain et al., 2012; Bairu & Kane, 2011) have demonstrated the general utility of plant tissue culture for conservation and propagation across various plant species, our study provides the specific protocol optimized for Clausena lenis. To our knowledge, no prior study has addressed in vitro micropropagation of C. lenis. Therefore, this work offers species-specific insights that directly supports the conservation and sustainable utilization of this medicinally valuable and under-researched plant. We revised this section of introduction at P2L52 as follows: “ While tissue culture protocols have been explored in other Clausena species, such as C. guillauminii [14] and C. harmandiana [15], to the best of our knowledge, there have been no published reports on in vitro propagation protocols specifically for C. lenis.”

Lines 53-55: What was obtained from these variations?

- We have revised the text to clarify what outcomes were obtained from the hormonal variations in the referenced studies. Specifically, we now highlight how these variations impacted the efficiency and quality of callus induction, shoot regeneration, and rooting in each species. This is at P2L58: “In Plumbago auriculata, adjusting auxin and cytokinin levels significantly improved both callus induction and shoot regeneration [17]. In Asparagus densiflorus, callus growth was more rapid on MS medium containing 5.4 μM pCPA and 4.4 μM BA compared to other combinations. BA alone was more effective than kinetin for shoot regeneration, and the addition of ancymidol to 0.4 μM BA significantly increased shoot numbers [18]. In Tectona grandis, specific hormonal combinations also enhanced shoot and root numbers [19]. These outcomes demonstrate how tailored hormone regimes can optimize specific developmental stages in plant tissue culture depending on species and explant type.”

Line 53: What do you mean by an efficient protocol? It may already exist.

avoids assuming the protocol is uniquely "efficient" and emphasizes that your contribution fills a gap in species-specific propagation knowledge.      

- We modified the text at P2L67 as follows “Developing a reproducible and well-characterized in vitro propagation protocol for C. lenis is important to provide a consistent supply of healthy plants with high propagation potential.”

Lines 60-61: This is known as an indirect organogenesis micropropagation protocol.

Since your study tested callus induction and shoot/root formation directly from explants, but did not regenerate shoots from callus tissue, it’s not an indirect organogenesis protocol.

- We revised the sentence to clarify this point at P2L70 as follows; “This study aims to determine effective disinfection strategies for C. lenis seeds, optimize protocols for callus induction, shoot and root formation directly from explants, and evaluate the survival rate of regenerated C. lenis plantlets once transferred to ex vitro conditions.”

Results

Table 1: Why wasn't NaCLO evaluated in 10 min?

- In our preliminary trials (data not shown), NaClO treatments at 10 minutes resulted in a very high rate of contamination and poor seed disinfection. Based on these observations, we proceeded with a 20-minute duration for all NaClO treatments to ensure a more rigorous and comparable evaluation of its disinfecting potential.

Why wasn't a second disinfection used in NaCLO?

- Prior to selecting the treatments presented in this study, we conducted preliminary tests using consecutive NaClO rinses at various concentrations and durations. However, these did not markedly improve the disinfection rate compared to a single-step treatment, and contamination levels remained unacceptably high. Therefore, HgCl₂ was included as a reference treatment due to its proven effectiveness.

Should alternatives to using HgCl2 be sought?

- Yes, we acknowledge the concerns regarding the toxicity and environmental risks associated with mercuric chloride. Although HgCl₂ proved highly effective in achieving 100% disinfection in this study, the development of safer, non-mercurial alternatives such as silver nanoparticles, plant-based antimicrobials, or optimized physical sterilization methods (e.g., UV or hot water treatments) should be explored in future work to establish more sustainable disinfection protocols.

Table 2: Why wasn't BAP vs. TDZ used?

- In this study, our focus was to initially screen a broad range of plant growth regulators to determine suitable combinations for callus induction in C. lenis. BA (BAP) was included at multiple concentrations based on its frequent use in woody plant micropropagation. TDZ, however, was tested only at 0.1 mg/L based on prior reports suggesting its high activity at low concentrations. We agree that a direct comparison between BA and TDZ across similar concentration ranges would have strengthened the assessment of their relative efficacy. This will be addressed in future studies aimed at fine-tuning hormonal combinations for optimal callus and organ induction.

Generating seed calluses could increase genetic variability.

- Indeed, callus cultures derived from seeds may exhibit increased genetic variability due to the inherent heterogeneity of seed-derived tissues and the potential for somaclonal variation during dedifferentiation. In our study, seed explants were selected primarily for their high responsiveness to 2,4-D-induced callus formation (100% induction), as shown in Table 2. While we acknowledge the possibility of genetic variation arising from seed-derived calluses, our primary aim was to establish an efficient protocol for callus initiation across explant types.

Table 3: Wasn't shoot regeneration based on the generated calluses? What was the purpose of generating calluses?

- In this study, shoot regeneration was not conducted from callus tissues. Instead, shoots were directly induced from explants (seeds, stem segments with nodes, and 1-week-old seedlings) cultured on MS media supplemented with different concentrations of cytokinins, as presented in Table 3. The purpose of generating calluses (Table 2) was to assess the responsiveness of different explant types to PGR induction for the potential of future applications, such as secondary metabolite production or indirect organogenesis. This is clarified in the revised manuscript to avoid any confusion at P4L129 as follows: “The impact of different PGRs on shoot induction from seeds, stem segments with four nodes and 1-week-old seedlings without roots is presented in Table 3.”

The results are interesting; however, they are part of a routine study in PTC. Techniques applied to all plant species to be micropropagated.

- We appreciate the reviewer’s comment and agree that many plant tissue culture (PTC) techniques are broadly applicable across species. However, this is a species-specific protocol for C. lenis, a medicinally and pharmacologically valuable but understudied species. Despite the routine nature of the methods, the study provides essential baseline data for future research into the conservation of C. lenis. We have emphasized the novelty and relevance of working with this species in the revised Introduction and Conclusion to better clarify the scientific value of the findings.

Discussion

Lines 182-185: According to which author?

- We have revised the sentence to include the appropriate citations and have added the corresponding references at P7L193 as follows: “The success of surface sterilization for seed and explants is influenced by various factors, including disinfectant concentration, duration of exposure, and the sensitivity of explants to chemical agents. Achieving high disinfection rates without compromising seed viability is essential for subsequent tissue culture steps [20,21].”

Lines 185-188, as can be seen, the use of HgCl2 in the cited works is not recent due to its current disuse.

Response:
- We acknowledge the reviewer’s observation regarding the reduced current use of HgCl₂ due to its toxicity and environmental concerns. However, in the absence of a more effective alternative for C. lenis seed sterilization, our preliminary trials with NaClO (including shorter durations) failed to eliminate contamination. Therefore, HgCl₂ was included as a benchmark treatment due to its historically high efficacy across multiple species. We have revised the discussion to clarify that although HgCl₂ is not widely recommended today, its inclusion in this study was for a protocol-establishing purpose, at P8L199 as follow: “Although HgCl₂ poses environmental and safety concerns, it was employed here due to the ineffectiveness of NaClO-based protocols in achieving contamination-free germination.”.

Lines 188-190, perhaps with two NaClO rinses, the contamination results would have decreased, and the use of HgCl2 would be avoided.

- Prior to selecting the treatments presented in this study, we conducted preliminary tests using consecutive NaClO rinses at various concentrations and durations. However, these did not markedly improve the disinfection rate compared to a single-step treatment, and contamination levels remained unacceptably high. Therefore, HgCl₂ was included as a reference treatment due to its proven effectiveness.

Lines 190-193, in this study, seed conservation and storage were not evaluated; this statement is incorrect because it was not proven.

- We have removed this statement regarding seed conservation and storage.

Lines 202-204, it would be worth discussing with another study where high doses of 2,4-D cause this.

- We have now included two additional references that report the effects of high concentrations of 2,4-D on tissue browning and growth inhibition in other plant species. This has been incorporated into the revised manuscript to strengthen the discussion in Lines 209–211.

Lines 209-210, according to whom?

- The references are in the followed sentences at P8L218-224 as here: “Furthermore, the use of different PGRs and their combinations significantly influences callus formation in various plant species. For example, Kitisripanya (2020) [13] found that 0.1 mg/L TDZ in combination with 1 mg/L NAA were effective in callus induction in Clausena harmandiana. Kanwar (2018) [35] found that a combination of 2,4-D and NAA resulted in high callus induction rates in Dianthus caryophyllus L. Kongkaew (2016) [36] reported that 2.0 mg/L 2,4-D and 0.5 mg/L BA, when combined, led to the highest callus in Hevea brasiliensis.”

Lines 221-225, in their study, BAP was not combined with TDZ; discuss comparable works on equal terms.
- In the revised manuscript, we have now clarified that while our study tested BA independently (without combination with TDZ), the cited studies include both single and combined cytokinin–auxin treatments. We have adjusted the wording to specifically refer to comparable studies using BA alone, while noting that other works employed different or combined PGR regimes, thereby maintaining a fair comparison. The revised text is at P L as follows: “In this study, 2.0 mg/L BA alone was sufficient to induce 100% shoot formation, with an average of 3.90 ± 0.31 shoots per explant. This supports earlier findings in Asparagus densiflorus and Terminalia bellerica using BA alone [18, 38]. While other studies used combined PGRs (e.g., BA with TDZ or auxins) [29, 39], our results suggest BA by itself can be effective for shoot induction in C. lenis.”.

Materials and Methods

Line 260, details should be given in the text so that the methods can be reproducible.

- We have added descriptions for the methods at P9L265 as follows: “First, they were immersed in 70% (v/v) ethanol for 1 minute, and then twelve disinfection treatments were evaluated, varying in chemical agents, concentrations, and exposure durations (Table 1). NaClO was tested at five concentrations (0.12%, 0.3%, 0.6%, 0.9%, and 1.2%) for 20 minutes as single-step treatments. Additionally, three concentrations (0.6%, 0.9%, and 1.2%) were tested in combination with a second disinfection step using 0.1% mercuric chloride (HgCl₂) for 5 minutes. Additional treatments included single-step disinfection with HgCl₂ at two concentrations (0.1% and 0.2%) for either 10 or 20 minutes. Each treatment involved 10 seeds (one per bottle), and disinfection and germination rates were recorded after 4 weeks of culture on MS medium.”

Lines 261-263: Was the culture medium sterilized? How?

- We have added descriptions for the methods at P9L274 as follows: “The seed coats were removed before placing the seeds on Murashige and Skoog (MS) medium supplemented with 3% (w/v) sucrose and 0.82% (w/v) agar, sterilized by autoclaved at 121°C for 15 minutes.”

Line 271: Where did the three sources of explants come from? From germinated seeds? Specify.

- We have added descriptions for the methods at P9L285 as follows: “Three types of sterile explants, including seeds, stem segments with four nodes (1.5–2.0 cm in length) taken from 1-month-old seedlings, and 1-week-old seedlings without roots, were obtained from germinated seeds of C. lenis and cultured on MS medium.”

Lines 274-276: What concentrations?

- We have added descriptions for the methods at P9L288 as follows: “N⁶-Benzyladenine (BA), 2,4-Dichlorophenoxyacetic acid (2,4-D), Thidiazuron (TDZ), and α-Naphthalene acetic acid (NAA) at concentrations ranging from 1.0–2.0 mg/L for BA, 0.5–1.0 mg/L for 2,4-D, 0.1 mg/L for TDZ, and 1.0–2.0 mg/L for NAA, either individually or in combination, as presented in Tables 2 and 3.”

Lines 278: Why were they evaluated at three months? And in other sections at six weeks?

- The evaluation for callus and shoot formation was performed after 3 months because earlier observations at 1–2 months did not yield consistent or clearly distinguishable responses in most treatments. In contrast, root formation was observable and well-developed within 6 weeks, which is why a shorter assessment period was used for that stage.

The materials and methods should be further detailed to make them reproducible.

- We have revised the Material and Methods as suggested above.

Conclusion

Line 310: Conducting a study for the first time does not justify publishing it.

Lines 318-320: To meet these objectives, a conservation strategy should have been developed.

The conclusion makes statements that were not verified in this study and should be improved.

- We appreciate the reviewer’s constructive feedback. We have revised the conclusion section following the comment from Reviewer 1. This resulted in a removal of unsupported claims and better reflect the results obtained from the study. The updated conclusion now focuses solely on the outcomes demonstrated through our experiments, including specific data related to disinfection efficiency, callus and shoot induction, rooting, and plantlet acclimatization.

Reviewer 3 Report

Comments and Suggestions for Authors

The manuscript provides a series of experiments examining the efficacy of sterilization techniques on disinfection and germination rates and the effect of various plant growth regulators on callus, shoot and root induction. The goal is establish a viable protocol for propagation of Clausena lenis Drake.

The authors provide a series of experimental designs to provide statistical rigor in evaluating which factors are important. The designs are not standard and the results can be improved by providing additional details to the reader. Some recommendations include:

In Table 1, the number of seeds used in each treatment should be identified. It is not clear if the 12 treatments are designed to understand the response (disinfection, germination) with time as a factor, in addition to disinfectant (bleach or Hg). The discussion (line 183) notes duration of exposure as a factor, but this is not changed in the case of bleach concentrations. Are there two factors or three being investigated? Are interactions evaluated? The "Experiment design and data analysis" mentions interactions (line 306). I doubt the design can provide any meaningful interpretation on interactions. In addition, Duncan's test is a post-ANOVA test - the results of the ANOVA also need to be displayed and discussed in the context of the experimental design and model. Which factors are qualitative and which are quantitative? Are the assumptions of normality and constant variance investigated?

The same general concerns hold for Tables 2-4. The post-ANOVA Duncan's test needs to be supported by the ANOVA results. Also, in Table 2, which are the factors and which are the responses? For instance, it would make perfect sense to interpret the data with 'seed', 'stem with nodes' and '1-week-old shoot' as (qualitative) factors in combination with PGRs and dosage level. Is this what was done? Otherwise, the experiment is run as an independent analysis for each potential factor just mentioned and comments on how seeds, stems or shoots are different are not able to be supported by any comparative statistic.

The data has many values for the responses at maxima (100%) or minima (0%) and not much needs to be inferred if the goal is 100%. The authors should comment on the limitations of their designs in interpreting outcomes - can they really measure interactions between factors? The authors should not describe outcomes as 'optimized' since no models beyond main effects and interactions seem to be used. The language is simply maximum.  

Author Response

The manuscript provides a series of experiments examining the efficacy of sterilization techniques on disinfection and germination rates and the effect of various plant growth regulators on callus, shoot and root induction. The goal is establish a viable protocol for propagation of Clausena lenis Drake.

The authors provide a series of experimental designs to provide statistical rigor in evaluating which factors are important. The designs are not standard and the results can be improved by providing additional details to the reader. Some recommendations include:

In Table 1, the number of seeds used in each treatment should be identified. It is not clear if the 12 treatments are designed to understand the response (disinfection, germination) with time as a factor, in addition to disinfectant (bleach or Hg). The discussion (line 183) notes duration of exposure as a factor, but this is not changed in the case of bleach concentrations. Are there two factors or three being investigated? Are interactions evaluated? The "Experiment design and data analysis" mentions interactions (line 306). I doubt the design can provide any meaningful interpretation on interactions. In addition, Duncan's test is a post-ANOVA test - the results of the ANOVA also need to be displayed and discussed in the context of the experimental design and model. Which factors are qualitative and which are quantitative? Are the assumptions of normality and constant variance investigated?

The same general concerns hold for Tables 2-4. The post-ANOVA Duncan's test needs to be supported by the ANOVA results. Also, in Table 2, which are the factors and which are the responses? For instance, it would make perfect sense to interpret the data with 'seed', 'stem with nodes' and '1-week-old shoot' as (qualitative) factors in combination with PGRs and dosage level. Is this what was done? Otherwise, the experiment is run as an independent analysis for each potential factor just mentioned and comments on how seeds, stems or shoots are different are not able to be supported by any comparative statistic.

  • We have clarified that the number of seeds used per treatment was 10, as now stated explicitly in the revised methods (at P9L292 and P10L305). Our experimental design involved a single-factor comparison (disinfectant treatment, hormone treatment, etc.) rather than factorial combinations. While interactions between factors were mentioned in the original version, we now recognize this was misleading given our design. The term has been removed, and the revised text now more accurately reflects the nature of our analyses.
  • Following your comment, we conducted an evaluation of the assumptions required for ANOVA, including normality and homogeneity of variances. The results indicated that the data from our four experiments (Tables 1–4) were not normally distributed. Therefore, ANOVA was deemed inappropriate for our dataset. Instead, we reanalyzed the data using the Kruskal–Wallis test, a non-parametric alternative suitable for comparing multiple independent groups when data distribution does not meet parametric assumptions. Post hoc pairwise comparisons were conducted using Fisher’s Least Significant Difference (LSD) test with Holm correction for multiple comparisons. This method allowed us to statistically confirm significant differences between treatment groups without relying on the assumption of normality.
  • The grouping results from the Kruskal–Wallis and post hoc analyses were consistent with our previously reported findings, reinforcing the validity of our conclusions. We have now clearly indicated the use of the Kruskal–Wallis test in the Tables 1–4, specifying the significance level as p < 0.001. Additionally, we revised the “Experiment Design and Data Analysis” section in the Materials and Methods to reflect the updated statistical approach.

The data has many values for the responses at maxima (100%) or minima (0%) and not much needs to be inferred if the goal is 100%. The authors should comment on the limitations of their designs in interpreting outcomes - can they really measure interactions between factors? The authors should not describe outcomes as 'optimized' since no models beyond main effects and interactions seem to be used. The language is simply maximum.

  • We acknowledge the limitation that our experimental design does not formally test for interactions between factors, and therefore we do not claim to have optimized the protocol in a statistical modeling sense. To reflect this, we have replaced the term "optimized" with "suitable" or “adjusted” throughout the manuscript to avoid overstatement.

Round 2

Reviewer 2 Report

Comments and Suggestions for Authors

The manuscript has been substantially improved, and the comments made have been corrected. The manuscript can be accepted in its current form.

Reviewer 3 Report

Comments and Suggestions for Authors

The authors have revised the manuscript and developed more robust statistical tests to support their conclusions.